# Repeat modules and N-linked glycans define structure and antigenicity of a critical enterotoxigenic *E. coli* adhesin

Zachary T. Berndsen[1¤a], Marjahan Akhtar[2], Mahima Thapa[3], Tim J. Vickers[2], Aaron Schmitz[3], Jonathan L. Torres[1], Sabyasachi Baboo[4], Pardeep Kumar[2¤b], Nazia Khatoon[2], Alaullah Sheikh[2], Melissa Hamrick[2], Jolene K. Diedrich[4], Salvador Martinez-Bartolome[4], Patrick T. Garrett[4], John R. Yates, III[4], Jackson S. Turner[3], Renee M. Laird[5¤c], Frédéric Poly[5], Chad K. Porter[6], Jeffrey Copps[1], Ali H. Ellebedy[3,7,8], Andrew B. Ward[1]*, James M. Fleckenstein[2,5,9]*

1 Department of Integrative Structural and Computational Biology, Scripps Research, La Jolla, California, United States of America, 2 Department of Medicine, Division of Infectious Diseases, Washington University in Saint Louis, School of Medicine. Saint Louis, Missouri, United States of America, 3 Department of Pathology and Immunology, Washington University in Saint Louis, School of Medicine, Saint Louis, Missouri, United States of America, 4 Department of Molecular Medicine, The Scripps Research Institute, La Jolla, California, United States of America, 5 Operationally Relevant Infections Department, Naval Medical Research Command (NMRC), Silver Spring, Maryland, United States of America, 6 Translational and Clinical Research Department, Naval Medical Research Command (NMRC), Silver Spring, Maryland, United States of America, 7 Center for Vaccines and Immunity to Microbial Pathogens, Washington University School of Medicine, St Louis, Missouri, United States of America, 8 The Andrew M. and Jane M. Bursky Center for Human Immunology and Immunotherapy Programs, Washington University School of Medicine, St Louis, Missouri, United States of America, 9 Medicine Service, Infectious Diseases, Veterans Affairs Health Care System, Saint Louis, Missouri, United States of America

¤a Current Address: Current affiliation: Department of Biochemistry, University of Missouri Columbia, Columbia, Missouri, United States of America.
¤b Current Address: Current affiliation: Pfizer, Saint Louis, Missouri, United States of America.
¤c Current Address: Current affiliation: Department of Diarrheal Disease Research, Walter Reed Army Institute of Research, Silver Springs, Maryland, United States of America.
* andrew@scripps.edu (ABW); jfleckenstein@wustl.edu (JMF)

**Data Availability Statement:** Structural data has been provided for EtpA in complex with monoclonal antibodies 1C08 and 1G05 in the Protein Data Bank under PDB IDs 8VBA, and 8VBB,

## Abstract

Enterotoxigenic *Escherichia coli* (ETEC) cause hundreds of millions of cases of infectious diarrhea annually, predominantly in children from low-middle income regions. Notably, in children, as well as volunteers challenged with ETEC, diarrheal severity is significantly increased in blood group A (bgA) individuals. EtpA, is a secreted glycoprotein adhesin that functions as a blood group A lectin to promote critical interactions between ETEC and blood group A glycans on intestinal epithelia for effective bacterial adhesion and toxin delivery. EtpA is highly immunogenic resulting in robust antibody responses following natural infection and experimental challenge of volunteers with ETEC. To understand how EtpA directs ETEC-blood group A interactions and stimulates adaptive immunity, we mutated EtpA, mapped its glycosylation by mass-spectrometry (MS), isolated polyclonal (pAbs) and monoclonal antibodies (mAbs) from vaccinated mice and ETEC-infected volunteers, and determined structures of antibody-EtpA complexes by cryo-electron microscopy. Both bgA and mAbs that inhibited EtpA-bgA interactions and ETEC adhesion, bound to the C-terminal repeat domain highlighting this region as crucial for ETEC pathogen-host interaction. MS

and in the Electron Microscopy Data Bank under EMD-43118, and EMD-43119, respectively.

**Funding:** This work was supported by National Institute of Allergy and Infectious Diseases (NIAID) of the National Institutes of Health (NIH) R01 AI089894, and R01 AI126887 to JMF and by funding from the Department of Veterans Affairs (5I01BX001469-05) to JMF. Research conducted by AS was also supported by National Institute of Allergy and Infectious Diseases of the National Institutes of Health under Award Number T32AI007172. The content is solely the responsibility of the authors and does not necessarily represent the official views of the National Institutes of Health, or the Department of Veterans Affairs. The funders had no role in study design, data collection and analysis, decision to publish, or preparation of the manuscript.

**Competing interests:** JMF is listed as the "Inventor" on U.S. patent 8323668 assigned to the University of Tennessee Research Foundation on December 4, 2012 that relates to use of EtpA related antigens in vaccine development. The other authors have declared that no competing interests exist.

analysis uncovered extensive and heterogeneous N-linked glycosylation of EtpA and cryo-EM structures revealed that mAbs directly engage these unique glycan containing epitopes. Finally, electron microscopy-based polyclonal epitope mapping revealed antibodies targeting numerous distinct epitopes on N and C-terminal domains, suggesting that EtpA vaccination generates responses against neutralizing and decoy regions of the molecule. Collectively, we anticipate that these data will inform our general understanding of pathogen-host glycan interactions and adaptive immunity relevant to rational vaccine subunit design.

## Author summary

Enterotoxigenic *E. coli* (ETEC), a leading cause of diarrhea disproportionately affecting young children in low-income regions, are a priority for vaccine development. Individuals possessing A blood-type are more susceptible to severe cholera-like disease. EtpA, a secreted, immunogenic, blood group A binding protein, is a current vaccine target antigen. Here, we determined the structure of EtpA in complex with protective as well as non-protective monoclonal antibodies targeting two different domains of the protein, pinpointing key regions involved in blood-group A antigen recognition and uncovering the mechanism of antibody-based protection. In addition, we show through mass-spectrometry that EtpA is extensively and heterogeneously glycosylated at surface-exposed asparagine residues by a promiscuous and low-fidelity glycosyltransferase, EtpC, and that this unique form of bacterial glycosylation is critical for to development of protective immune responses. Lastly, polyclonal antibodies from vaccinated mice as well as monoclonal antibodies obtained from ETEC-infected volunteers revealed that the highly antigenic surface of EtpA exhibits both protective and non-protective epitopes. These results greatly expand our understanding of ETEC pathogenesis, and the immune responses elicited by these common infections, providing valuable information to aid in the rational design and testing of subunit vaccines.

## Introduction

Enterotoxigenic Escherichia coli (ETEC) are diarrheal pathogens defined by their production of heat-labile (LT) and heat-stable (ST) enterotoxins [1]. ETEC, an exceedingly common cause of infectious diarrhea in areas where clean water and sanitation remain limited, accounts for hundreds of millions of cases of acute diarrheal illness each year [2]. In addition, these pathogens are a leading cause of more severe diarrhea and death [3,4] among young children of low-income regions and are associated with long-term sequelae including poor growth [5–9] and malnutrition [10–13].

Given the persistent and pervasive impacts of ETEC infections, these pathogens have remained a high priority for vaccine development [14–16]. Efforts to identify novel surface-expressed molecules that might be targeted in ETEC vaccine development led to the identification of the plasmid-borne etpBAC two-partner secretion (TPS) locus responsible for export of EtpA, an extracellular adhesin [17]. At a minimum, TPS loci are comprised of a transmembrane polypeptide-transport-associated (POTRA) [18] domain (TpsB) protein responsible for secretion of a cognate (TpsA) exoprotein. The corresponding components of the etpBAC

locus include EtpB the transmembrane protein required for secretion of the extracellular EtpA adhesin, as well as EtpC, a glycosyltransferase responsible for glycosylation of EtpA [17]. All three genes are required for optimal secretion of EtpA. The EtpA molecule is typically heavily glycosylated and etpC mutants exhibit dramatically reduced production of EtpA, as well as altered tropism for target epithelial cells, suggesting that glycosylation of EtpA may be important for proper folding and function of the adhesin [17].

Once secreted, the high molecular weight (~170 kDa) EtpA glycoprotein serves as a unique molecular bridge between the bacteria and intestinal mucosal surfaces [19], essential to pathogen-host interactions required for delivery of both LT [20,21] and ST [22]. On host epithelia, EtpA binds to N-acetylgalactosamine (GalNAc) residues on enterocyte surfaces as well as secreted mucins including MUC2, interactions that are critical for efficient adhesion, toxin delivery, and intestinal colonization [23]. EtpA preferentially engages GalNAc as the terminal sugar of human A blood group presented on enterocytes. Importantly, volunteers challenged with the EtpA-producing H10407 strain of ETEC were significantly more likely to develop moderate-severe diarrhea if they were blood group A [24], recapitulating earlier observations that young bgA+ children in Bangladesh were more likely to develop diarrhea with ETEC infection [9].

In exploring the utility of EtpA as a potential vaccine antigen, studies to date have demonstrated that the etpBAC locus is highly conserved across ETEC from geographically disparate origins [25–30], is immunogenic following natural [27,30] and human experimental challenge [31,32] infections, and that immunization with recombinant EtpA (rEtpA) affords considerable protection against intestinal colonization [19,21,27,29,33–37]. In addition, EtpA expression by ETEC strains is strongly associated with the development of diarrheal illness in young children, while antibodies against EtpA are associated with protection [30]. Despite enthusiasm for targeting EtpA in next-generation ETEC vaccines [38–40], relatively little is known about its structure, antigenicity, or the mechanisms by which antibodies targeting this molecule mediate protection.

Several structures of truncated N-terminal (TPS) domains required for secretion of TpsA molecules [41–45] as well as a single full-length TpsA exoprotein [46], have been solved by X-ray crystallography. The highly homologous TPS domain structures all adopt a similar fold, specifically, an extended 3-sided β-helix. These N-terminal TPS domains may be followed by a series of repeat modules, with some such as HxuA [46] containing short extra-helical loops or motifs thought to contribute to function [47]. The N-terminal secretion domain of EtpA was previously shown to be sufficient for export as well as binding to flagella [19], while the function of a series of four C-terminal repeats was unknown.

Here we provide the complete structure of EtpA determined by cryo-EM, both alone and complexed to anti-EtpA monoclonal antibodies produced by vaccination, as well as high-resolution mass-spectrometry (MS) analysis of EtpA glycosylation. These data define regions of the molecule that are required for activity and are targets for antibody neutralization, provide a detailed profile of its extensive antigenic glycosylation, and show that alterations in glycosylation can impact antibody function. We identified a diverse set of epitopes from polyclonal sera of vaccinated mice, indicating that EtpA possesses a large and variable immunogenic surface with numerous potential neutralizing and decoy epitopes. We anticipate that the elucidation of the antigenic structure of this important virulence factor will afford insights into molecular correlates of protection and help guide further development of EtpA and similar proteins as vaccine immunogens.

## Results

### EtpA repeat regions direct blood group A binding on target host cells

Like many bacterial adhesins, EtpA is a lectin, or a carbohydrate binding protein [23]. Similar to another TpsA protein, filamentous hemagglutinin of *Bordetella pertussis* [48,49], EtpA also possesses the ability to agglutinate erythrocytes. Hemagglutination activity of lectins typically arises when these molecules possess two or more carbohydrate binding sites permitting cross-linking of cells [50]. While individual interactions may be of low affinity, high avidity can be achieved through tandem repetition of lectin-binding regions [51–53]. To determine whether the C-terminal region of EtpA, comprised of 4 repeat modules (Fig 1A), was involved in blood group A glycan recognition, we first examined a truncated version of recombinant EtpA (rEtpA$_{1-1086}$, Fig 1A), lacking the full complement of repeat modules. We found that while the truncated molecule was efficiently secreted, it was incapable of binding efficiently to blood group A glycans on the surface of intestinal epithelial cells (Fig 1B), on solid substrates (Fig 1C), or erythrocytes (Fig 1D) suggesting that the repeats act in concert to engage target carbohydrates.

### Antibodies targeting EtpA repeats interrupt bgA binding and ETEC adhesion

To identify potential protective epitopes on EtpA, we examined the capacity of anti-EtpA monoclonal antibodies (mAb) to impair EtpA binding to A blood group glycans, and interrupt pathogen-host interactions. Two mAbs isolated from rEtpA-vaccinated mice, 1G05 and 1C08, both bound to EtpA with high affinity, (S1A Fig), but recognized distinct epitopes on EtpA (S1B–S1C Fig). The 1G05 mAb, which recognized the CTR domain (S1D Fig) significantly inhibited interactions with blood group A (Fig 1E–1F) and impaired bacterial adhesion (Fig 1G). Conversely, 1C08, which recognized the NTS domain (S1D Fig), exhibited no demonstrable impact on EtpA binding to target blood group A molecules or ETEC pathogen-host interactions. This pattern was also observed in monoclonals isolated from volunteers challenged with ETEC H10407, with the three monoclonals that recognized the NTS domain (Fig 1H) failing to inhibit EtpA-bgA interactions (Fig 1I). Conversely the single monoclonal (1F09) that bound the CTR significantly inhibited EtpA interactions with BgA. Collectively, these data indicate that the C-terminal repeat region of EtpA is essential to ETEC virulence, and that antibodies targeting this region can effectively inhibit interactions with the host.

### The EtpA adhesin forms an elongated β-helix

To obtain the complete structure of EtpA and gain further insight into the differential activity of the 1G05 and 1C08 mAbs, we performed cryo-EM analysis of rEtpA bound to the fragment antigen binding domains (Fab) of both mAbs, resulting in reconstructions of 4 and 3.3 Å-resolution for the 1G05 and 1C08 bound complexes, respectively (S2 and S3 Figs). Similar to other TpsA exoproteins, the mature EtpA molecule forms an elongated and slightly twisted 3-sided parallel β-helix ~25 nm in length (Fig 2A) that can be divided into amino-terminal secretion (NTS- residues 66:640) domain, and carboxy-terminal repeat (CTR–residues 641:1534) domain. The NTS domain, also referred to as the TPS domain, is highly conserved among TpsA proteins and is required for interactions with the polypeptide transport-associated (POTRA) domains of the outer membrane β-barrel transporter (TpsB) [41,54]. The NTS domain contains the only extra-helical inserts present on EtpA (Fig 2B), which fold back over the exterior of the main β-helix. Unlike the closely related pectate lyase protein, which binds carbohydrate residues in the pocket created by its inserts [55], those of the EtpA NTS domain

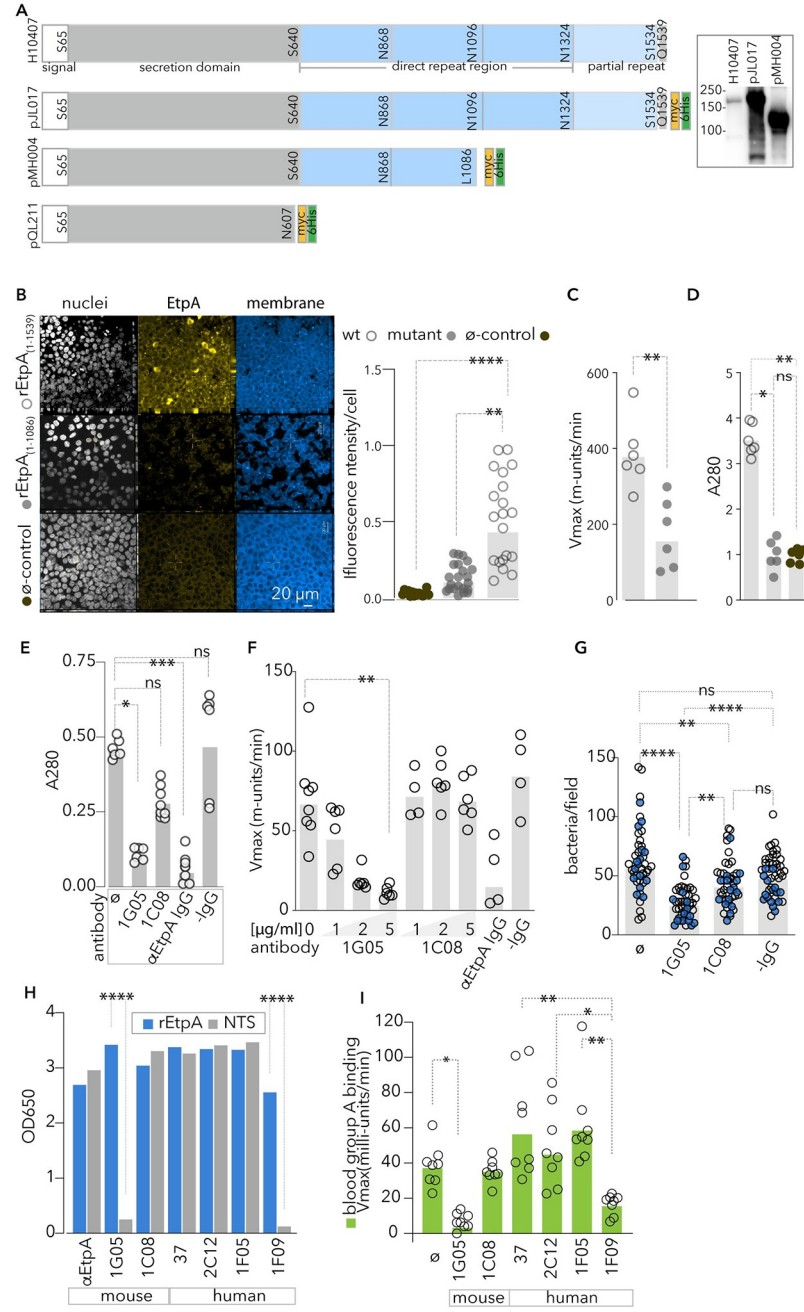

**Fig 1. The repeat region of EtpA directs critical interactions with A blood group glycans.** A. Schematic depicts molecular organization of the EtpA molecule from ETEC H10407 (top), recombinant EtpA encoded on pJY017, and the truncated recombinant antigens EtpA (1–1086) and the NTS domain (1–607) encoded on pMH004, and pQL211, respectively. Inset: anti-EtpA immunoblot of TCA-precipitated culture supernatants from H10407 wild type strain, and recombinant Top10 strains jf1696 and jf5090 carrying plasmids pJL017 and pMH004, respectively. B. Binding of full-length and truncated, mutant EtpA molecules to blood group A-expressing HT-29 cells. Nuclei, (white) are shown at left; center column: protein binding (yellow); right: cell membranes (CellMask Plasma Membrane Stain ThermoFisher). Shown at left are representative fields quantified in graph at right from n = 20 replicate fields from two independent experiments. Bars indicate geometric mean fluorescence intensity per cell (****≤0.0001, ** = 0.0015 by Kruskal-Wallis nonparametric testing). C. Kinetic ELISA data reflect binding of full-length and truncated EtpA to blood group A ** = 0.0043 (Mann-Whitney, two-tailed). D. Blood group A1 erythrocyte (A1-RBC) pull-down assay with full-length and truncated EtpA. * = 0.0147, **0.0073 by Kruskal-Wallis. Results in C, D represent combination of technical duplicates from three independent experiments. E. Inhibition of EtpA-A1-RBC interactions with anti-EtpA monoclonal antibodies recognizing the repeat (mAb 1G05) and secretion (mAb 1G08) domains compared to anti-

EtpA mouse polyclonal IgG, and negative IgG isotype control (-IgG) * = 0.0103, *** = 0.0002. F. mAb inhibition of blood group A-EtpA interaction ** = 0.0084 (Kruskal-Wallis). G. Anti-EtpA mAbs inhibit ETEC bacterial adhesion. Data reflect replicate experiments (closed and open symbols represent respective experiments) (n = 45 fields total) and the impact of anti-EtpA mAbs on ETEC adhesion to target blood group A expressing HT-29 cells ****<0.0001, **<0.001(Kruskal-Wallis). Grey bars throughout represent geometric mean values. H. mAb recognition of full-length rEtpA (blue bars) and NTS of EtpA (grey bars) in end-point ELISA. Data represent geometric mean of ≥ 6 technical replicates combined from 2 experimental replicates. ****<0.0001 by ANOVA. I. mAb inhibition of EtpA binding to human A blood group in kinetic ELISA assay. N = 8 technical replicates from 2 independent experiments (Ø = no antibody control; *<0.05, **<0.005, by Kruskal-Wallis).

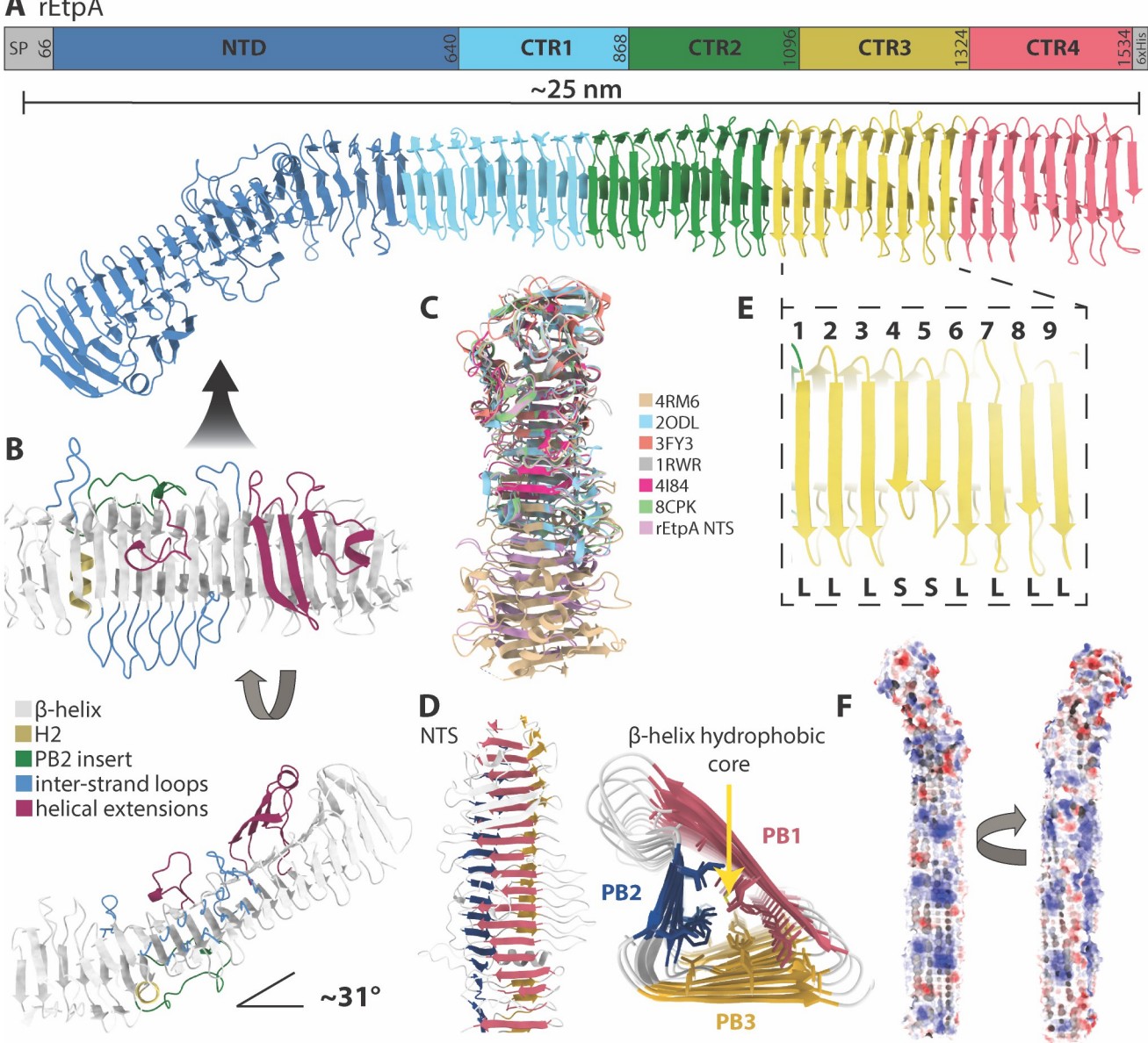

**Fig 2. The cryo-EM structure of rEtpA.** A. Color coded sequence diagram and atomic model of EtpA derived from cryo-EM maps. B. Top-down and side views of the N-terminal secretion domain (NTS, a.k.a. TPS) of EtpA with various features color coded. C. Structural alignment of previously published NTS domains from related TpsA proteins along with their corresponding PDB identifiers. D. Top-down view of the EtpA NTS domain and and CTR domain viewed looking down the core of the β-helix, both color coded by the 3 by parallel β-sheets, PB1, PB2, and PB3. E. Zoomed in view of one CTR showing the 9 β-strands designated as either long (L) or short (S). F. Surface representation of the EtpA structure colored by Coulombic potential.

do not form solvent-accessible binding pockets, and their structural and/or functional importance are unknown. The NTS domain also contains the only two α-helices in the structure, H1, and H2, the latter of which creates a ~31˚ kink in the otherwise straight β-helix (Fig 2A–2B). A similar alpha helix-induced kink was observed between the NTS and functional C-terminal domains in the crystal structure of HxuA from *Haemophilus influenzae* [46] (PDB 4RM6) potentially enhancing flexibility between the two domains. The structure of the EtpA NTS domain determined here aligns closely with the recently characterized crystal structure of this region (PDB 8CPK) [56], is similar to the published crystal structures of the N-terminal domains of related TpsA proteins including HxuA (PDB 4I84, [46] PDB 4RM6) [42], the HMW1 (PDB 2ODL) [43] adhesin from *Haemophilus influenzae*, hemolysin A from *P. mirabilis* (PDB 3FY3) [44], and filamentous hemagglutinin from *Bordetella pertussis* (PDB 1RWR) [41] as illustrated in the structure-based alignment (Fig 2C).

The CTR domain contains three 228 residue repeats followed by a 219-residue partial repeat forming an unbroken 3-sided β-helix (Fig 2A). Following previously established convention, the three parallel β-sheets forming the sides of the helix are referred to as PB1, PB2, and PB3 (Fig 2D). PB1 and PB2 are both continuous β-sheets composed of ~54 strands, while PB3 is split into two parts by H2 (Fig 2A and 2B). Within the CTR domain, PB1 is the widest with strands that are 5–7 residues long, followed by PB3 with strands that are 4–5 residues long, then PB2 with strands that are only 2–3 residues long (Fig 2D). Each CTR is composed of 9 β-strands per side (Fig 2E) and separating each strand are loops which form the edges of the helix, with the loops separating PB1 and PB2 being the longest (Fig 2D). The fourth and fifth strand of each repeat on PB1 are shorter than the others (Fig 2E), however, the reason for this minor asymmetry is not apparent. The interior of the β-helix is composed almost entirely of closely packed hydrophobic residues which contribute to the stability of the helical structure (Fig 2D), while the exterior has an abundance of polar and charged residues (Fig 2F). Altogether, the lack of extra-helical extensions or apparent binding clefts on the CTR suggest that bgA must interact directly with the β-helix of this domain.

## Glycosylation of EtpA by EtpC, a promiscuous low-fidelity N-linked glycosyltransferase

Perhaps the most striking feature of the EtpA structure is its unique and extensive surface glycosylation. In addition to the *etpBAC* operon, other TPS loci from *Yersinia*, and *Burkholderia spp.* appear to encode glycosyltransferases related to HMW1C of *H. influenzae* [57–59]. The structure of the EtpC glycosyltransferase predicted by AlphaFold2 [60] shows high structural similarity (pruned Ca-RMSD = 1.1Å; all residue Ca-RMSD = 3.7Å) to the crystal structure of the closely related HMW1C glycosyltransferase from *Actinobacillus pleuropneumoniae* [61] (S4 Fig), a functional homolog to the HMW1C glycosyltransferase of *H. Influenzae* which shares ~40% sequence identity, and 56% similarity to EtpC. The HMW1 adhesin glycosylated by the HMW1C enzyme in *H. influenzae* exhibits a unique glycosylation profile consisting of asparagine-linked (N-linked) mono- and di-hexose glycans appearing predominately at canonical N-X-S/T sequences (where X is any amino acid expect proline), with a single modification of a non-canonical asparagine (Asn) residue [62]. Additional analysis identified the hexose residues as primarily glucose and sometimes galactose [58], with the HMW1 glycosyltransferase catalyzing the formation of both the Asn-hexose and hexose-hexose linkages. HMW1C exhibited no apparent selection for modification of distinct sequons with either mono or dihexose sugars. Glycosylation by HMWC1 likely stabilizes the HMW1A adhesin and is required to tether this molecule to the surface of *H. influenzae*. Similarly, EtpC is required for efficient secretion and function of the EtpA exoprotein adhesin [17]. Presently

however, neither the glycosylation profile conferred by EtpC or its precise impact on pathogen-host interactions are understood.

To identify the location and type of glycan modifications on rEtpA we employed high- resolution site-specific mass spectrometry (MS) [63]. We analyzed potential glycosylation at 166 out of the 196 non-tandem Asn residues (we did not detect peptides associated with 30 Asn residues), of which 96 adhere to the canonical (N-X-S/T) N-linked glycosylation sequon, and we found evidence for hexose modification at 133 sites, 94 of which meet the stricter criteria of $\geq$ 25% occupancy (Fig 3A). Based on the occupancy across all potential N-glycosylation sites (PNGS), mature EtpA would have on average 61 glycans per molecule, meaning that ~1 in every 24 residues (~4%) of the EtpA exoprotein harbors an N-glycan modification. Comparatively, $\leq$ 2% of HMW1A [64] and 1.7% of the SARS-CoV2 spike protein [65] are glycosylated. Among sites with the highest occupancy ($\geq$ 75%) only 4 out of the 29 are non-canonical Asn residues, (S5–S7 Figs), suggesting that like HMW1A, Asn residues within canonical sequons are glycosylated with higher fidelity. All 4 of these non-canonical glycosylation sites fall within the same repeating sequence/structural motif located on the last β-strand of each CTR in PB1. Though the majority of PNGS were found to be occupied with monohexose, dihexose was observed at 83 sites, but only 4 of those sites, N744, N972, N1200 and N1428, were found to have $\geq$ 50% dihexose, again all belonging to a common repeating structural motif located on the second short β-strand of each CTR in PB1 (Fig 3A and 3B). Altogether these data suggest that the surface glycan coat of EtpA is both dense and variable.

When viewed in a structural context, we see that the confirmed N-linked glycosylation sites (NGS) on EtpA are asymmetrically distributed across the protein (Fig 3B–3D). PNGS within the NTS domain are glycosylated with significantly higher fidelity and specificity (for the canonical N-linked glycosylation sequon) than Asn residues within the CTR domain. For example, the NTS accounts for ~39% of the protein (and ~39% of Asn residues), however, 22 out of the 33 PNGS (~67%) without any detected glycan modifications were within the NTS, and 19 of those were within the first ~400 residues. Further, the NTS domain contains the two NGS (N290 and N349) with the highest occupancies ($\geq$ 95%). These data suggest that the EtpA sequence as well as structural determinants may dictate glycosylation by EtpC

## N-linked glycan clustering and intramolecular interactions on rEtpA

As illustrated by mapping the occupancy-weighted local glycan density onto the protein surface (Fig 3C), the NGS are more evenly dispersed across the CTRs, but significantly more abundant on the PB1 face of the β-helix (Fig 3C and 3D). Our cryo-EM maps confirm the location of many of these hexose modifications, as shown in the refined model (Fig 3D). Of the glycosylated residues up through CTR2 where the cryo-EM map permitted identification (84 >0%, 60 $\geq$25%, 43 $\geq$50%, 19 $\geq$75% occupancy), we were able to model hexose residues at 39. Although previous analytical studies of HMW1A reveal a mixture of glucose and galactose residues, we are unable to differentiate between glucose and galactose with MS alone, so all hexose residues were modeled as glucose for consistency (Fig 3D). We did not observe clear map density for dihexose modifications at any NGS.

Given the extensive glycosylation of EtpA and the poor yields obtained in prior attempts to express the exoprotein without EtpC, we questioned whether the glycans might contribute to stabilization, folding and secretion of the protein. The stabilizing effect of N-linked glycans is at least partially mediated through favorable interactions with neighboring amino acid side chains, often via stacking with aromatic residues or hydrogen bonding with polar residues. This stabilizing interaction almost always involves the core *N*-acetylglucosamine, which in the case of EtpA would be equivalent to the N-linked hexose residue [66,67].

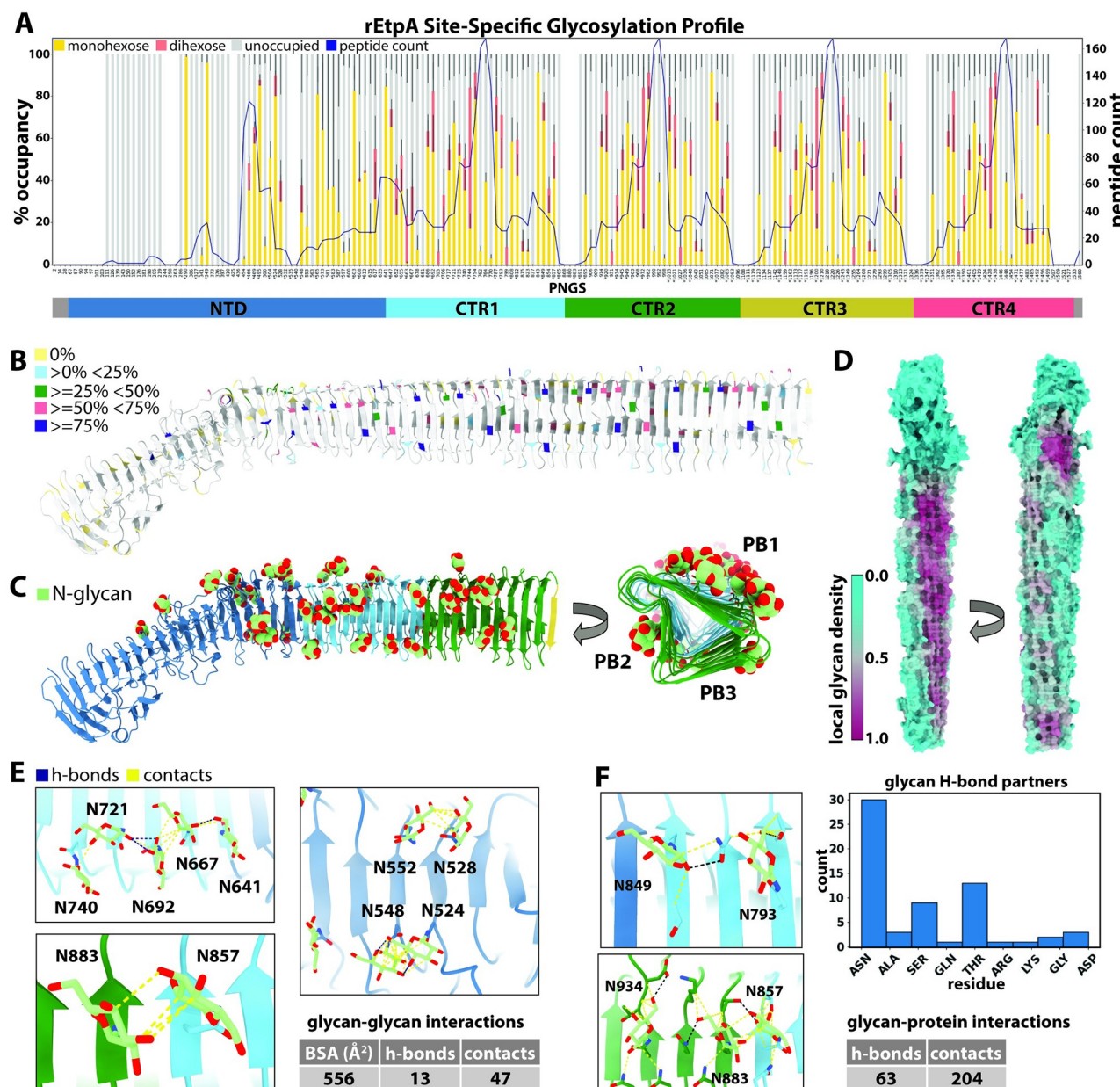

**Fig 3. N-linked glycosylation of rEtpA.** A. Site-specific mass-spectrometry data for rEtpA showing the % occupancy by monohexose, or dihexose, or no glycan (unoccupied) at all 196 asparagine residues analyzed along with the peptide count (right axis and blue line—number of peptides detected in the experiment) and corresponding domain diagram. The * under certain residues indicates a canonical PNGS sequon site. B. Structure of rEtpA with potential N glycosylation sites (PNGS) colored by % occupancy. C. Surface representation of rEtpA structure colored by local glycan density (20Å radius). D. Refined model of rEtpA colored by domain showing all modeled N-linked glucose residues (through CTR2) viewed from the side and looking down the core of the β-helix. E. Analysis of glycan-glycan interactions and (F) glycan-protein interactions from the refined model of rEtpA showing hydrogen bonds (blue) and contacts (yellow) along with summary tables and a histogram of all glycan-protein residue hydrogen bonding partners. BSA = Buried Surface Area.

Although we did not find statistical enrichment of any aromatic residues around the NGS (S8A–S8C Fig), our structure did reveal numerous glycan-glycan, as well as glycan-amino acid interactions with other residue types (Fig 3E and 3F). Further, the NGS on EtpA have a tendency group into local clusters. For example, of the 11 glycans modeled on PB1, 5 of

them, including 1 more from the final strand of the NTS domain, are located immediately adjacent to each other on the first residue of each β-strand and can be seen to form a chain of inter-glycan interactions (Fig 3E, top left). Other clusters of 2 or more glycans are observed throughout the structure on both the NTS and CTR domains (Fig 3E; bottom left, right). In total, there are 13 potential glycan-glycan hydrogen bonds and 47 contacts captured in our structure, resulting in 556Å of buried glycan surface area. In addition, we identified 63 potential glycan-amino acid hydrogen bonds, with the majority involving N residues as well as T and S (Fig 3F). Although individually weak, these numerous small interactions likely contribute collectively to overall EtpA stability.

Finally, we sought to determine the glycosylation profile of native full-length EtpA from ETEC strain H10407. Despite inherent difficulties in protein purification and low peptide detection in MS, we were able to map the glycosylation state at 48 PNGS of the native protein, with variable occupancy similar to that exhibited by rEtpA. (S9 Fig).

## Molecular interactions with the mAbs 1C08 and 1G05

Our cryo-EM maps confirm that 1C08 monoclonal binds the NTS domain (Figs 4A and S3) while the neutralizing monoclonal 1G05 binds the CTR region (Figs 4A and S2). The 1G05 epitope encompasses PB1 and the inter-strand loops between PB1 and PB2 and is located at the interface of the repeat domains, allowing for the binding of up to 3 Fabs per molecule (Figs 4A and S2). The 1C08 epitope encompasses PB3 and the long inter-strand loops between PB3 and PB1, just before the H2 helix (Fig 4A). 1G05 and 1C08 share ~88% sequence identity and have similar structures (Cα-RMSD = 0.77Å, Fig 4B and 4C), despite being derived from different germline genes (S10 Fig). Both Fabs utilize their heavy chain (HC) complementarity determining region (CDR) loops extensively, while 1C08 also makes numerous contacts with its light chain (LC) CDR loops, and their binding interfaces bury 864Å$^2$ and 1275Å$^2$ of surface area, respectively (Fig 4D). The larger buried surface areas and more extensive interactions of 1C08 are consistent with the kinetics data showing tighter binding to rEtpA. The weaker binding of 1G05 is compensated for by avidity affects arising from the possible binding of both Fab arms of a single IgG to each EtpA molecule. Consistent with the different binding modes, analysis of somatic hypermutation from the predicted unmutated common ancestor (UCA) germline germline sequence using ARMADiLLO [68] shows that 1G05 harbors 11 mutations in its HC as opposed to 7 for 1C08, while 1C08 has a more mutated LC with 6 mutations compared to 1G05 with only 3 (S10 Fig).

A defining feature of both epitopes is the presence of a single centrally located N-linked hexose residue, N349 and N849 for the 1C08 and 1G05 epitopes, respectively (Fig 4E). Intriguingly, N349 has the second highest occupancy of any site as determined by MS (~96%) and N849 is in the top 17% of sites by occupancy. This could be taken as evidence that antibodies targeting PNGS with higher occupancy are enriched during affinity maturation, or conversely, that heterogeneity in glycosylation is being exploited as a mechanism of immune evasion. 1C08 targets an epitope with relatively low local glycan density in the NTS, while 1G05 targets an epitope with high local glycan density, however, 1G05 is oriented perpendicular to the long axis of the β-helix and utilizes a smaller HC dominant binding interface such that it positions its LC away from the heavily glycosylated surface of PB1, thus avoiding all but a single glycan residue. The local density analysis also reveals that it would be difficult for antibodies to target an entirely glycan-free epitope on the CTR domain, while there is ample glycan-free surface area for potential antibody binding on the NTD.

Lastly, to determine the extent to which these glycans contribute to the affinity of mAb 1G05 for epitopes within the CTR, we mutated EtpA asparagine residues at N849, N1077, and

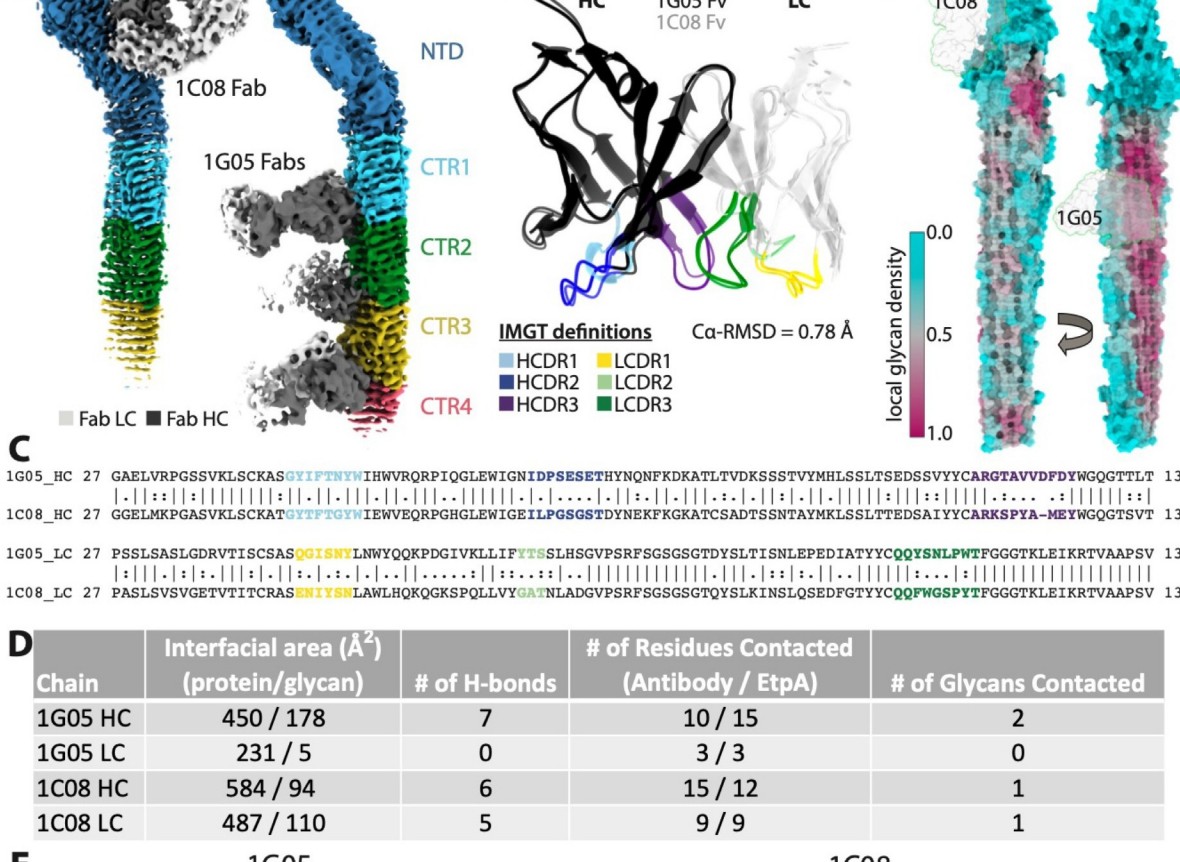

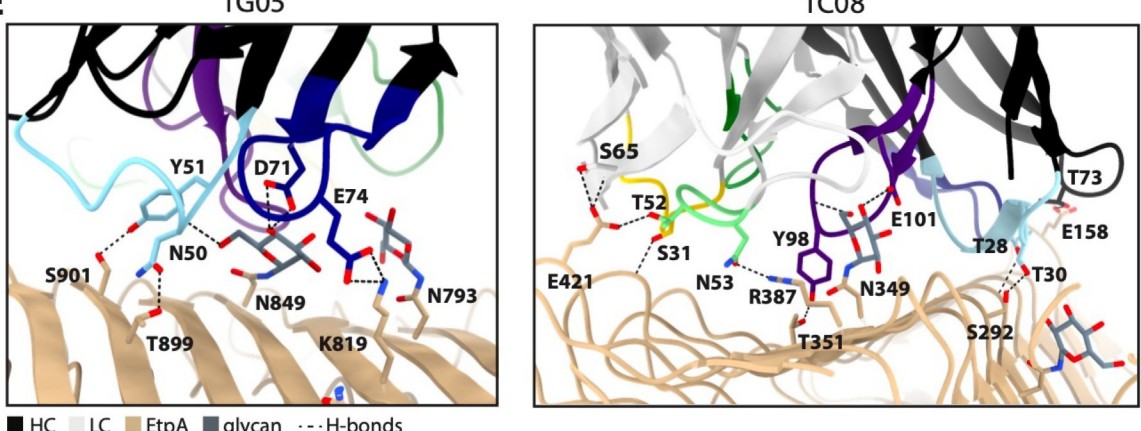

**Fig 4. Molecular interactions between rEtpA and mAbs 1C08 and 1G05.** A. Fabs Cryo-EM density maps of rEtpA in complex with 1C08 and 1G05 Fabs colored by domain. B. Structrual alignment of mAb Fc domains colored by heavy chain (HC) and light chain (LC) complementary-determining-region loops (CDRL) along with a Ca-RMSD. C. Pairwise sequence alignment of both Fv domains with CDRLs color coded. D. Table summarizing intermolecular interactions between rEtpA and mAbs calculated from PDBePISA. E. Intermolecular interactions between rEtpA and mAbs. F. Fabs 1 C08 and 1G05 projected onto glycan density maps of EtpA.

N1305 to alanine. Affinity of the 1G05 mAb for the mutant protein was significantly diminished while binding of 1C08 was unimpaired (S11A and S11B Fig). Importantly however, these mutations within the CTR did not impact interaction with target A blood group glycans (S11C Fig).

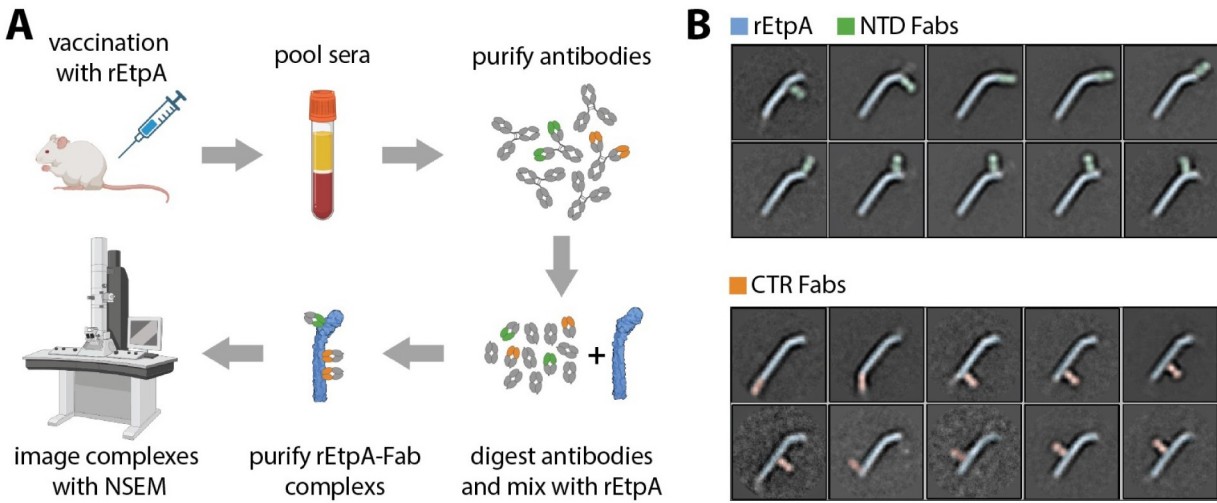

**Fig 5. EMPEM of pooled sera isolated from rEtpA vaccinated mice.** A. Simplified schematic of the EMPEM workflow (Created with BioRender.com). B. 2-dimensional class averages of rEtpA in complex with polyclonal Fabs showing all unique epitopes identified with EMPEM.

## Electron Microscopy Polyclonal Epitope Mapping (EMPEM) of sera from rEtpA-vaccinated mice

Earlier studies have demonstrated that vaccination with rEtpA affords significant protection against intestinal colonization by ETEC [21,36,37]. To further understand the potential immunogenic landscape of rEtpA we performed negative-stain electron microscopy (NSEM)-based polyclonal epitope mapping of sera from mice vaccinated with rEtpA (Figs 5A and S12). We found that polyclonal antibodies target a variety of epitopes on both the CTR and NTS domains of rEtpA (Fig 5B). With NSEM images of Fabs bound to rEtpA informed by the high-resolution cryo-EM structures, we were able to confidently identify antibodies bound at either the N- or C-terminal domains and estimate the total number of unique epitopes on each. We found that 5–6 and 3–4 unique epitopes are targeted on the NTS and CTR domains, respectively, suggesting that the polyclonal response to EtpA likely generates a variety of both neutralizing and decoy epitopes.

## Discussion

Although ETEC is an extraordinarily common cause of diarrheal morbidity in LMICs, there is presently no licensed vaccine to protect against these pathogens. To date, virtually all vaccine development efforts for ETEC have focused on a subset of heterogeneous canonical antigens known as colonization factors (CFs). To date 29 antigenically distinct CFs have been identified, potentially confounding the development of a broadly protective vaccine [69]. However, recent studies have suggested that other antigens including the EtpA adhesin are important for ETEC molecular pathogenesis, are recognized following both experimental human challenge as well as natural infections and are more highly conserved within the ETEC pathovar [26]. The studies outlined here provide a detailed structure of this complex extracellular glycoprotein adhesin molecule and offer further insight into the nature of its interactions with human host intestinal epithelia.

Clinical presentations associated with ETEC range from mildly symptomatic illness to severe diarrhea indistinguishable from cholera. Indeed, the early identification of the ETEC

pathovar resulted from investigating cases of *Vibrio cholerae*- negative cholera [70]. However, in contrast to *Vibrio cholerae* infections which tend to be more severe in blood group O subjects, A blood group not O, is associated with more severe ETEC illness. Notably, toxin delivery by ETEC requires intimate cell contact [20]. EtpA, a blood group A lectin originally identified in H10407, an ETEC isolate from a case of severe cholera-like diarrhea [71], accelerates binding and intoxication of blood group A epithelia [24]. These studies provided a molecular basis for the enhanced disease severity among A blood group individuals observed in both young children in endemic areas [9] as well as adult volunteers challenged [24] with H10407.

The present studies further define the nature of EtpA-mediated interactions with A blood group glycans. EtpA-targeting of blood group A glycans, which have N-acetylgalactosamine as their terminal sugar moiety, involves repeat modules comprising the C-terminal 2/3rds of the molecule. Interestingly, tandem repeats of carbohydrate binding modules (CBMs) [72–74] including those identified in other bacterial virulence molecules achieve tight binding through multivalent interactions with target host glycans [74]. The identification of epitopes critical for EtpA host interaction within the heavily glycosylated CBM repeats may have significant implications for the ETEC vaccine design. Multi-epitope fusion vaccines have been heralded as a potential strategy to achieve valency sufficient to protect against these diverse pathogens. The present studies suggest, however, that similar to HIV [75], a thorough understanding of antigenic structure, including glycosylation profiles, combined with precise identification of protective epitopes is essential to inform rational design of immunogens [76] that afford broad protection against these common pathogens.

The characterization of EtpC as a low-fidelity N-linked glycosyltransferase has important implications for understanding the immune response to these pathogens during infection and vaccination. Comparison of EtpA sequences from disparate geographic origins collected over time has revealed little amino acid variation despite the significant immunogenicity of this molecule [26]. The glycan-centered epitopes of mAbs characterized here may indicate that ETEC exploit glycosylation as a potential immune evasion mechanism. Many enveloped viruses hijack eukaryotic host cellular machinery to decorate their surface fusion proteins with "self" N-linked glycans facilitating immune evasion [75,77]. However, unlike EtpA, these viral fusion proteins can tolerate extensive mutation, allowing for PNGS to be added or removed readily. Thus, the EtpA adhesin may have evolved with its own low-fidelity glycosyltransferase under selective pressure to shield important and highly conserved functional regions of the molecule by generating a variety of N-linked glycan-profiles in immune evasion. The present studies are limited by difficulty inherent in mapping the glycosylation profile of the native protein, making it difficult to completely assess the potential impact of the two plasmid expression system [78] employed in production of the recombinant protein on antigenicity.

Negative stain EMPEM reveals that rEtpA presents a large immunogenic surface, with multiple potential epitopes on both the NTD and CTR domain, and analysis of anti-EtpA mAbs to date indicate that only some of these antibodies, particularly those targeting the CTR domain are likely to afford protection. Given the promise of recombinant subunit-based vaccines based on these glycosylated adhesins, it will be vitally important to ensure that such vaccine antigens present native-like epitopes with the proper glycosylation.

## Materials and methods

### Ethics statement

All procedures involving animals were performed in accordance with guidelines of Institutional Animal Care and Use Committee of Washington University in Saint Louis (protocol number 21–0053). The human study protocol was approved by the Naval Medical Research

Command Institutional Review Board in compliance with all applicable federal regulations governing the protection of human subjects. Formal written consent was previously obtained from all volunteer subjects. Use of de-identified specimens from volunteers was approved by the Washington University in Saint Louis Institutional Review Board (Protocol number 201110126).

## Molecular cloning of EtpA mutants

To construct a plasmid encoding EtpA$_{1-1806}$, a Tn7-based GPS4 transprimer-mutagenized (NEB) *etpA* genes [19] with insertions in the C-terminal repeat region of *etpA* were amplified from the indicated template plasmid (S1 Table) with primers jf051716.1 and jf082718.1 (S2 Table), to permit In-Fusion (TaKaRa) cloning of the resulting *etpBA* amplicon bearing a truncated *etpA* gene into pBAD/myc-His B digested with *NcoI/HindIII*, placing the EtpA truncation in-frame with the C-terminal polyhistidine tag. Plasmid pQL211 encoding the EtpB and amino terminal secretion domain of EtpA (EtpA$_{1-607}$) was generated from pJL017 (S1 Table) with primers jf042114.3/jf042114.4 (S2 Table).

The triple mutant EtpA-N849A-N1077A-N1305A was created by excision of the highly repetitive C-terminal region from the *etpA* gene by digestion of pJL017 with *KflI* and *Hind*III, then the insertion of three overlapping gene blocks f1-N849A, f2-N1077A, and f4-N1305A (IDT) by HiFi assembly (NEB). When inserted into the digested plasmid, these gene blocks created a synthetic gene with the three desired mutations. The resulting plasmid, pTV005 was verified by long-read nanopore sequencing (plasmidsaurus).

## Purification of recombinant EtpA and EtpA mutants

EtpA and subclones were purified as previously described [78]. Briefly, 75 ml of Terrific Broth containing ampicillin (100 μl/ml) and chloramphenicol (15 μg/ml) with 0.2% (w/v) glucose was inoculated with frozen glycerol stock of jf1696, jf3013, or jf5090, and grown overnight at 37 degrees C, 225 rpm. 5 ml of overnight growth was used to inoculate 2-liter flasks each containing 500 ml fresh media, and grown at 37 degrees C, 225 rpm to OD ~0.6 Cultures were then induced with 0.0002% arabinose (5 μl 20% arabinose/flask) x 4–5 hours @ 37 degrees C, 225 rpm. Cultures were then centrifuges at 8,000 rpm x 10 min, and supernatant stored overnight at 4 degrees C. After filtration through 0.2 μm filters, the filtrate was concentrated via tangential flow (Pellicon 2 Biomax 100 kDa MWCO) to a final volume of ~100 ml and applied to a 10 ml His-Trap column, then washed with 5 column volumes of binding buffer (50 mM PO4 pH 7.5 300 mM NaCl). Protein was eluted over a gradient of 50 mM PO4, pH 7.5, 300 mM NaCl, 1 M imidazole, fractions collected and analyzed by SDS-PAGE. Fractions with protein of interest were pooled and dialyzed vs 10 mM MES pH 6, 100 mM NaCl, then concentrated to final concentration of ~1 mg/ml.

For purification of native EtpA (nEtpA) from H10407, the flagellin null mutant (jf3099) was grown in 4 L LB media and the culture supernatant obtained and concentrated as before. The concentrate was then exchanged into 50 mM MES pH 6, 300 mM NaCl, 1 mM EDTA and proteins precipitated by addition of ammonium sulfate to 80% saturation. The resulting precipitate was recovered by centrifugation at 9,820 x g for 10 min, and the pellet redissolved in 5 ml PBS. This solution was then concentrated to 1 ml using a 50 K cutoff spin concentrator (Millipore). The sample was applied to a HiLoad 16/600 Superdex 200 gel filtration column equilibrated in PBS, and proteins separated at a flow rate of 1 ml min-$^{1}$. Fractions containing nEtpA were identified by SDS-PAGE, pooled, and concentrated to produce the final preparation.

## EtpA biotinylation

Primary amines of recombinant EtpA were biotinylated with EZ-Link Sulfo-NHS-LC-LC-Biotin (ThermoFisher 21388)) for 30 minutes at room temperature. The reaction was quenched with Tris 100 mM, pH 8.0, then reactants dialyzed to remove excess biotin.

## Mouse immunization and plasmablast sorting

Two female C57BL/6J mice (Jackson Laboratories) were immunized intramuscularly and boosted nine weeks later with 15 µg EtpA emulsified in PBS and AddaVax (InvivoGen). Draining iliac and inguinal lymph nodes were collected four days after boosting and single cell suspensions were prepared for plasmablast sorting. Cells were stained for 30 min on ice with CD138-BV421 (281–2, 1:200), IgD-FITC (11-26c.2a, 1:100), CD19-PE (1D3, 1:200), CD38-PE-Cy7 (90, 1:200), Fas-APC (SA367H8, 1:400), CD3-APC-Cy7 (17A2, 1:100), and Zombie Aqua (Biolegend) diluted in PBS supplemented with 2% FBS and 2mM EDTA. Cells were washed twice and single PBs (CD138+ CD38lo CD19+/lo CD3- live singlet lymphocytes) were sorted using a FACSAria II into 96-well plates containing 2 µL Lysis Buffer (Clontech) supplemented with 1 U/µL rNase inhibitor (NEB) and immediately frozen on dry ice.

## Plasmablast isolation from volunteers experimentally challenged with ETEC

Peripheral blood mononuclear cells obtained from volunteers challenged with ETEC strain H10407 [79] were obtained from archived specimens maintained at the National Medical Research Command, Silver Spring, Maryland. Cells were incubated for 10 min on ice with FcX (Biolegend), then stained for 30 min on ice with biotinylated rEtpA in PBS supplemented with 2% FBS and 2mM EDTA. Cells were washed twice and stained for 30 min on ice with CD4-Spark UV 387 (SK3, 1:200), CD8-Spark UV 387 (SK1, 1:100), CD14-Spark UV 387 (63D3, 1:200), CD20-Pacific Blue (2H7, 1:400), IgD-BV785 (IA6-2, 1:200), CD19-FITC (HIB19, 1:100), CD71-PE (CY1G4, 1:400), CXCR5-PE-Dazzle594 (J252D4, 1:50), CD38-PE-Fire810 (S17015F, 1:400), streptavidin-APC, and Zombie NIR (all Biolegend) diluted in PBS supplemented with 2% FBS and 2mM EDTA. Cells were washed twice and single rEtpA-binding PBs (rEtpA$^+$ CXCR5$^{low}$ CD71$^+$ CD20$^{low}$ CD38$^+$ IgD$^{low}$ CD19$^{+/int}$ CD4$^-$ CD8$^-$ CD14$^-$ live singlet lymphocytes) were sorted using a Bigfoot Spectral Cell Sorter (ThermoFisher Scientific) into 96-well plates containing 2 µL Lysis Buffer (Clontech) supplemented with 1 U/µL rNase inhibitor (NEB) and immediately frozen on dry ice.

## Monoclonal antibody isolation and purification

Antibodies were cloned as previously described [80,81]. Briefly, VH and Vκ genes were amplified by reverse transcriptase-polymerase chain reaction (RT-PCR) and nested PCR from single-sorted plasmablasts using cocktails of primers specific for IgG, IgM/A, Igκ, and Igλ using first round and nested primer sets [80–83] and then sequenced. Clonally related cells were identified by the same length and composition of IGHV, IGHJ and heavy-chain CDR3 and shared somatic hypermutation at the nucleotide level. To generate recombinant antibodies, heavy chain V-D-J and light chain V-J fragments were PCR-amplified from 1st round PCR products with mouse variable gene forward primers and joining gene reverse primers having 5' extensions for cloning by Gibson assembly as previously described [81,84] and were cloned into the pABVec6W human IgG1 antibody expression vector [85] in frame with either human IgG or IgK constant domain. Plasmids were co-transfected at a 1:2 heavy to light chain ratio into Expi293F cells using the Expifectamine 293 Expression Kit (Thermo Fisher), and

antibodies were purified with protein A agarose (Invitrogen). From 94 sorted cells, 49 rearranged IGHV sequences were recovered, of which 31 were clonally distinct. 7 clonally distinct mAbs were generated, of which 5 bound EtpA.

## EtpA human A blood group interaction studies

**Erythrocyte pull down.** Erythrocyte pull-down studies were performed with 6His-tagged EtpA linked to cobalt-coated magnetic beads (Dynabeads, Invitrogen 10103D) and blood group A1 red blood cells (RBC) obtained from Immucor (0002345). Briefly, beads were re-suspended by vortexing, washed with 1 ml of PBS, then re-suspended in 100 μl of PBS and 100 μl of EtpA-6His at a final concentration of 1 mg/ml, and incubated at room temperature on a rotary mixer (~10 rpm) x 20 minutes. Beads were separated on a magnet, the supernatant removed, and then washed in 1 ml of PBS. After re-suspension in 100 μl of fresh PBS, EtpA-linked beads were maintained on ice during RBC preparation. A1 RBCs were re-suspended by inverting the tube of cells, and 1 ml of the suspension was transferred to a 1.5 ml microfuge tube on ice. RBCs were spun at 1000 rpm at 4 degrees C x 1 minute. Supernatant was then removed and RBCs were washed 4 times in low ionic strength solution (LISS)–- 1.75 g/L NaCl, 18 g/L glycine, 0.01%(w/v) sodium azide, 11.3 ml of 150 mM $KH_2PO_4$ stock, 8.7 ml of 150 mM $Na_2HPO_4$ stock, final pH 7.0.

**EtpA-blood group A ELISA.** Blood group A conjugated to BSA (Dextra NGP6305) was dissolved in PBS containing 0.02%(w/v) azide to a final concentration of 0.5 mg/ml, and stored at 4 degrees C prior to use. A working solution of bgA-BSA conjugate was prepared in carbonate buffer, pH 9.6 at a final concentration of 1 μg/ml. 100 μl of the solution was then used to coat each well of ELISA strips (Corning 2580) overnight at 4 degrees C. Wells were then washed 3 x with 200 μl of PBS containing 0.02% Tween-20 (PBS-T), then blocked with 100 μl of 1% BSA in PBS-T at 37 degrees C for 1 hour. Coated wells were then probed with 100 μl of EtpA-biotin (10 μg/ml in PBS-T-1% BSA)/ well for 2 hours at room temperature. Plates were washed 5 x with of PBS-T, and incubated with avidin-HRP conjugate (BioRad 1706528, diluted 1:10,000 in 1% BSA in PBS-T) for 1 hour at room temperature, then washed again 4x with PBS-T. Plates were developed with freshly prepared room-temperature HRP substrate (TMB-(3,5,",")-tetramethylbenzidine)-2 component reagent (seracare 5120–0053), and read kinetically at 650 nm (blue).

**ETEC adhesion to blood group A intestinal epithelia.** HT-29 cells (ATCC HTB-38) [24] which express blood group A glycans were propagated as previously described in McCoy's-5A medium (Gibco, Life Technologies) supplemented with 10% bovine serum albumin. Cells were grown to confluence in 96-well plates and incubated at 37degrees C, 5% CO2 for one week prior to use. Adhesion assays were performed as previously described [19] using mid-log phase bacterial cultures. After 30 minutes monolayers were washed 3 x with pre-warmed media, then treated with 0.1% Triton-X-100 in PBS for five minutes. Dilutions of the resulting lysates were plated onto Luria agar and bacterial adherence expressed as the percentage of the original inoculum recovered.

**Confocal laser scanning microscopy.** HT-29 cells seeded onto poly-L-lysine treated glass coverslips were incubated in 24 well plates at 5% CO2, 37˚C to confluence. Bacteria were added at a multiplicity of infection of ~1:100 and incubated for 1 hour prior to fixation. Cell-Mask deep red plasma membrane stain (Thermo Fisher Scientific, C10046) (1:2,000) and DAPI (1:6000) were used to stain cells and A blood group was detected with mouse monoclonal antibody Z2A (Santa Cruz sc-69951) against human A blood group antigen, followed by AlexaFluor 647-conjugated goat anti-mouse IgM heavy chain (Molecular Probes, A21238). Confocal images were acquired using a Nikon Eclipse Ti2 inverted microscope. ETEC H10407

(serotype O78) were imaged using polyclonal antisera (Rabbit) supplied by the Penn State E. coli Reference Center, followed by cross-absorbed goat anti-rabbit IgG (H\&\L) conjugated to either AlexaFluor 488 or 594 fluorophores (Invitrogen).

## Electron microscopy

**Cyro-EM sample preparation.** 450 μg of purified rEtpA was combined with 12x molar excess of 1G05 or 1C08 Fabs and incubated overnight at 4˚C. Complexes were then purified via SEC with a HiLoad 16/600 Superdex 200 pg column (GE Healthcare) on an AKTA Pure 25M system (Cytiva) using MES pH 6 as the running buffer. SEC peaks corresponding to rEtpA:Fab complexes were pooled and concentrated with Amicon 10 kDa concentrators to a final concentration of ~1 mg/ml. 3 μL of each complex was briefly incubated with lauryl maltose neopentyl glycol (LMNG; Anatrace) to a final concentration of 0.005 mM and then deposited on glow discharged Quantifoil Cu 1.2/1.3–300 mesh grids and plunge frozen using a Thermo Fisher Vitrobot Mark IV at 4˚C, 100% humidity, blot force 1, 10 second wait time, and a blot time of 4–7 seconds.

**Cryo-EM- - data collection.** EM micrographs of rEtpA in complex with 1G05 or 1C08 Fabs were collected on a Thermo Fisher Glacios cryo-electron microscope operated at 200keV and 96k x magnification (pixel size = 0.725-angstrom), equipped with a Thermo Fisher Falcon 4 direct electron detector, and operated with Thermo Fisher EPU 2 software. For the rEtpA + Fab 1G05 complex, ~9,000 micrographs were collected each with a total dose of ~47 e-/angstrom-squared, fractionated over 40 frames, at a target defocus range of -1.5 to -0.5 micrometer–- For a complete description of imaging conditions and data statistics see S3 Table.

**Cryo-EM- - data processing.** Both Cryo-EM datasets were processed with CryoSparc v2 [86] using the following standard workflow. Movie micrographs were aligned, and dose weighted with patch motion correction and the contrast transfer function (CTF) was fit to each micrograph using the patch-based CTF algorithm followed by manual curation to remove micrographs with poor CTF fit parameters and ice thickness. Blob picking was performed on a subset of curated micrographs followed by particle extraction, 2D-classification, and subset selection. Particles associated with good 2D classes were used to train a Topaz neural network [87] which was then used to pick particles from the entire curated dataset. These particles were then extracted and multiple rounds of 2D classification followed by subset selection were performed. Next, multiple rounds of reference free *Ab Initio* 3D classification followed by subset selection were performed to identify particle stacks that refine well and to separate particles with different stoichiometric ratios of bound Fabs, ranging from 0–3 Fabs in the case of 1G05 and 0–1 Fabs for 1C08. These subsets were then refined separately using non-uniform 3D refinement [88] with per-particle and global CTF refinement enabled [89]. A final round of focused 3D refinement was performed with masks around Fab epitopes.

**Cryo-EM- - model building and refinement.** AlphaFold2 [90] implemented through ColabFold [91] was used to generate the starting EtpA model for refinement. Due to the large size of EtpA the sequence was broken down into ~700 residue fragments with ~100 residue overlaps (S13 Fig) and each fragment was aligned in UCSF Chimera [92] using the 100 residue overlaps followed by removal of repeated residues and model combination into a single PDB file. Fab models were generated with SABPred [93] and added to the complete rEtpA AlphaFold2 model in UCSF Chimera and combined into a single PDB file. This combined file was then refined using ROSETTA [94] asking for ~300 models. Each model was scored with MolProbity [95] and EMRinger [96] and the model with the top combined score was selected. Next, N-linked glucose residues were added manually using COOT [97] and a ligand restraint file was generated in Phenix [98] using eLBOW [99]. Glucose residues were covalently linked

to 39 PNGS up through the end of CTR2, after which map resolution diminished. The cryo-EM maps also revealed probable hexose density at 4 sites that were not observed to be glycosylated by MS, namely N540, N883, N552, and N685, the first two of which occur within peptides not detected by MS, while the last two are incorrectly assigned tandem Ns. This glycosylated model was then refined with Phenix real-space refinement and model quality was assessed with the software mentioned above. N-linked glycans were validated with Privateer [100]. If adjustments were necessary, they were performed manually in COOT and followed up with additional rounds of real-space refinement in Phenix.

**Structural analysis.** Molecular graphics for images and molecular contact calculations were performed with UCSF ChimeraX, developed by the Resource for Biocomputing, Visualization, and Informatics at the University of California, San Francisco, with support from National Institutes of Health R01-GM129325 and the Office of Cyber Infrastructure and Computational Biology, National Institute of Allergy and Infectious Diseases [101]. Epitope-Paratope interactions were analyzed with PDBePISA [102].

## Negative Stain EM polyclonal epitope mapping

Sera obtained from CD-1 mice vaccinated intranasally (IN) with recombinant full-length EtpA -myc-His (rEtpA) were pooled and used to prepare polyclonal IgG as described previously [103,104]. Mice were vaccinated IN with 20 μg of rEtpA adjuvanted with 1 μg of dmLT [105] on days 0, 14, 21 as previously described [21,36,37], followed by terminal bleed on day 35. 5 ml of pooled mouse sera was then used to purify IgG using Protein A Sepharose resin (GE Healthcare) and digested for using papain-agarose resin (Thermo Fisher Scientific). Fc and undigested IgG were removed through with Protein A Sepharose resin using 0.2 ml packed resin per 1 mg of IgG. Fab samples were concentrated, and buffer exchanged to TBS using Amicon ultrafiltration units with a 10 kDa cutoff (EMD Millipore Sigma) and mixed in excess with rEtpA and allowed to incubate overnight at 4°C. rEtpA:polyclonal Fab complexes were then purified via SEC, concentrated to ~0.01mg/ml and prepared for imaging as described previously using uranyl formate stain [103]. EM micrographs were collected on an FEI Spirit microscope operating at 120keV and controlled with Leginon software [106]. Single-particle negative stain image processing was performed with Relion/4.0 software [107] as described previously [103]. For figures, features in 2D-class averages corresponding to Fabs were identified by eye and false-coloring was applied for clarity.

## Site-specific glycosylation analysis of rEtpA

**Sample preparation.** Recombinant EtpA glycoprotein was exchanged to water using Microcon Ultracel PL-10 centrifugal filter. Glycoprotein was reduced with 5 mM tris(2-carboxyethyl) phosphine hydrochloride (TCEP-HCl) and alkylated with 10 mM 2-Chloroacetamide in 100 mM ammonium acetate for 20 min at room temperature (RT, 24°C). Glycoprotein was digested with 1:25 Proteinase K (PK) for 30 min at 37°C or 1:20 trypsin for 16 h at 37°C. Proteases (PK/trypsin) were denatured by incubating at 90°C for 15 min, further diluted in buffer A, subsequently analyzed by LC-MS/MS.

Proteinase K and trypsin treatment: recombinant EtpA glycoprotein was exchanged to water using Microcon Ultracel PL-10 centrifugal filter. Glycoprotein was reduced with 5 mM tris(2-carboxyethyl) phosphine hydrochloride (TCEP-HCl) and alkylated with 10 mM 2-Chloroacetamide in 100 mM ammonium acetate for 20 min at room temperature (RT, 24°C). Glycoprotein was digested with 1:25 Proteinase K (PK) for 30 min at 37°C or 1:20 trypsin for 16 h at 37°C. Proteases (PK/trypsin) were denatured by incubating at 90°C for 15 min, further diluted in buffer A, subsequently analyzed by LC-MS/MS

**LC-MS/MS.** Samples were analyzed on an Orbitrap Eclipse Tribrid mass spectrometer. Samples were injected directly onto a 25 cm, 100 μm ID column packed with BEH 1.7 μm C18 resin. Samples were separated at a flow rate of 300 nL/min on an EASY-nLC 1200 UHPLC. Buffers A and B were 0.1% formic acid in 5% and 80% acetonitrile, respectively. The following gradient was used: 1–25% B over 100 min, an increase to 40% B over 20 min, an increase to 90% B over another 10 min and held for 10 min at 90% B for a total run time of 140 min. Column was re-equilibrated with buffer A prior to the injection of sample. Peptides were eluted from the tip of the column and nanosprayed directly into the mass spectrometer by application of 2.8 kV at the back of the column. The mass spectrometer was operated in a data dependent mode. Full MS1 scans were collected in the Orbitrap at 120,000 resolution. The cycle time was set to 3 s, and within this 3 s the most abundant ions per scan were selected for HCD MS/MS at 35 NCE. Dynamic exclusion was enabled with exclusion duration of 60 s and singly charged ions were excluded.

**Data processing.** Protein and peptide identification were done with Integrated Proteomics Pipeline (IP2). Tandem mass spectra were extracted from raw files using RawConverter [108] and searched with ProLuCID [109] against a database comprising UniProt reviewed (Swiss-Prot) proteome for *Escherichia coli* K12 (UP000000625) with amino acid sequence for EtpA (NCBI: WP001080112.1) containing C-term MYC and 6xHis-tag, and a list of general protein contaminants. The search space included no cleavage-specificity for PK, and half-tryptic specificity with unlimited missed cleavages for trypsin. Carbamidomethylation (+57.02146 C) was considered a static modification. Monohexose (+162.052824 N) and Dihexose (+324.105647 N) were considered differential modifications and a maximum of 3 differential modifications were allowed per peptide. Data was searched with 50 ppm precursor ion tolerance and 500 ppm fragment ion tolerance. Identified proteins were filtered using DTASelect2 [110] and utilizing a target-decoy database search strategy to limit the false discovery rate to 1%, at the spectrum level [111]. A minimum of 1 peptide per protein, and no tryptic end per peptide for PK and 1 tryptic end per peptide for trypsin were required and precursor delta mass cut-off was fixed at 10 ppm. Statistical models for peptide mass modification (modstat) were applied (for trypsin, trypstat statistics used). Census2 [112] label-free analysis was performed based on the precursor peak area, with a 10 ppm precursor mass tolerance and 0.1 min retention time tolerance. Data analysis using GlycoMSQuant [113] was implemented to automate the analysis. GlycoMSQuant summed precursor peak areas across replicates, discarded peptides without NGS, discarded misidentified peptides when N-glycan remnant-mass modifications were localized to non-NGS asparagines and corrected/fixed N-glycan mislocalization where appropriate. Census output files were modified to accommodate differential mass modification notations in GlycoMSQuant. The GlycoMSQuant algorithm was modified to accommodate quantification at all asparagines without implementing sequon-specific definition of NGS.

**Sequenced-based analysis.** Statistical analysis of PNGS flanking residue frequencies were performed with custom python scripts available at (https://github.com/ZTBioPhysics/EtpA-Glycosylation-Analysis.git). Briefly, for every residue in the input list, the identity of the amino acids 2 residues immediately upstream and downstream were determined and the overall frequencies for each amino acid type were calculated. To calculate p-values a permutation test was used. 1000 replicates of randomly generated residues drawn from the EtpA sequence without replacement of the same length as the original list were generated and analyzed in the same manner. The number of those replicates with higher or lower frequencies than the input list for each amino acid type were used to calculate empirical p-values. $P < 0.001$ was labeled with"**", $p < 0.01$ with"*", $p < 0.05$ with"", and $p > 0.05$ with"n".

**Residue proximity analysis.** Statistical analysis of neighboring residues was performed with custom python scripts available at (https://github.com/ZTBioPhysics/

EtpA-Glycosylation-Analysis.git). Briefly, for every residue in the input list, the identity of all the amino acids within a given distance (here 6Å) were determined and the overall frequencies for each amino acid type were calculated. Emperical p-values were calculated with the same type of permutation test as described above.

**Local glycan density analysis.** The local glycan density calculations were performed with custom python scripts available at (https://github.com/ZTBioPhysics/EtpA-Glycosylation-Analysis.git). Briefly, the EtpA atomic model was divided into 3 separate structures corresponding to each parallel β-sheet, PB1, PB2, and PB3, so that local proximity measurements would not include residues on opposite sides of the β-helix. For each residue in each of the 3 split structures, the number of PNGS withing a given distance were calculated for every residue then normalized between 0 and 1 and the B-factor column in each PDB file was replaced with this value. The three structures were then combined and the UCSFChimeraX function 'render-by-attribute' was used to color the surface representation of rEtpA with the local glycan density value. Considering the size of a Fab variable domain is ~40Å width-wise, a 20Å probe radius was used to simulate the local density that would be encountered by a Fab binding to the surface.

## AlphaFold2 modeling of EtpC

The sequence for EtpC (GenBank AAX13510.1) was fed to AlphaFold2 implemented via ColabFOLD [114] with standard parameters. The top ranking model was compared to the crystal structure of HMW1C (PDBID:3Q3E) in UCSF Chimera.

## Biolayer interferometry

The binding affinities of 1C08 and 1G05 Fabs to rEtpA were measured using an Octet Red 96 instrument (Forte Bio) and Ni-NTA biosensors. The entire assay was performed in 1X kinetics buffer (1X PBS pH 7.4 + 0.02% Tween-20 + 0.1% bovine serum albumin). There was a total of five steps in the assay: baseline 1 (60 sec), loading (180 sec or 1 nm threshold, whichever came first), baseline 2 (60 sec), association (180 sec), and dissociation (600 sec). The rEtpA antigen was loaded on the Ni-NTA biosensors at 25 ug/ml and the fabs were tested at 7 different concentrations, starting at 500 nM to 7.81 nM (2-fold dilutions). Using the Octet Analysis software (ForteBio), the data were reference subtracted and curves were fit with a 1:1 binding model. $K_D$, and On and off-rates were determined with a global fit.

## Disclaimer

The views expressed in this article reflect the results of research conducted by the authors and do not necessarily reflect the official policy or position of the Department of the Navy, Department of Defense, nor the United States Government. R.L., F.P. and C.P. are federal employees of the United States government. This work was prepared as part of official duties. Title 17 U.S.C. 105 provides that 'copyright protection under this title is not available for any work of the United States Government.' Title 17 U.S.C. 101 defines a U.S. Government work as work prepared by a military service member or employee of the U.S. Government as part of that person's official duties.

## Supporting information

**S1 Fig. Anti-EtpA mAbs exhibit distinct affinities to unique regions of the antigen. A.** Biolayer inferometry (Octet) studies 1C08 and 1G05 Fabs binding to rEtpA. **B.** mAb 1C08 does not compete for binding with 1G05. Shown are kinetic ELISA data indicating binding of

bioinylated 1G05 mAb in the presence of unlabeled 1G05 (blue), 1C08 (green) or alone (open circle). **C.** 1G05 and 1C08 recognize EtpA but compete for different binding sites. **D.** 1G05 recognizes the repeat region of EtpA while 1C08 binds the N-terminal secretion domain. Data include n = 6 technical replicates and are representative of three independent experiments. pAb = polyclonal anti-EtpA antibody. Comparisons by Kruskal-Wallis ***≤0.001, **≤0.01, *<0.05.
(TIF)

**S2 Fig. Cryo-EM data processing workflow for rEtpA-1C08 complex.** Simplified cryo-EM data processing workflow including A representative example of an aligned and dose-weighted micrograph (lowpass filtered to 5Å), 2D and 3D class averages, and particle counts at each step. **B.** Angular distribution and Fourier shell correlation plots for the final 3D reconstruction along with the final particle count. **C.** Sharpened map colored by domain. **D.** Map colored by local resolution estimate. **E.** View of the map-model fit in the epitope/paratope region.
(TIF)

**S3 Fig. Cryo-EM data analysis for rEtpA-1G05 complex.** Simplified cryo-EM data processing workflow including representative (A) aligned and dose-weighted micrograph (lowpass filtered to 5Å), 2D and 3D class averages, and particle counts at each step. B. Angular distribution and Fourier shell correlation plots for the final 3D reconstruction along with the final particle count. C. Sharpened map colored by domain. D. Map colored by local resolution estimate. E. View of the map-model fit in the epitope/paratope region.
(TIF)

**S4 Fig. AlphaFold2 prediction of EtpC structure and structure-based alignment with the HMW1C crystal structure.** A. Predicted structure of EtpC colored with a rainbow color mapping from the N-terminus to C-terminus and (B) by prediction LDDT confidence store. C. Multiple sequence alignment coverage. D. Predicted LDDT score by residue position. E. Predicted alignment error matricies. F. Structure-based alignment of predicted EtpC structure with the crystal structure of HMW1C (PDBID:3Q3E).
(TIF)

**S5 Fig. Statistical analysis of residues flanking canonical sequon PNGS.** Statistical analysis of amino acid type frequencies at the 4 sites immediately upstream and downstream of each canonical sequon PNGS broken down by occupancy percentage as determined by mass-spectrometry. Blue bars are the average frequencies for the input residue list and orange bars are average frequencies for 1000 permutation samples with error bars and statistical significance measurements broken down by minor significance (*), median significance (**), and high significance (***) along with corresponding p-values.
(TIF)

**S6 Fig. Statistical analysis of residues flanking non-canonical sequon PNGS.** Same as in S5 Fig, but for all non-canonical sequon PNGS.
(TIF)

**S7 Fig. rEtpA structure depicting sequon and non-sequon PNGS sites.** A. rEtpA structure with the 8 sequon PNGS sites with 0% occupancy colored red. **B.** rEtpA structure showing all non-sequon PNGS with ≥ 50% occupancy colored red. **C.** Bar plot showing the frequency of secondary structure types at the PNGS and the two residues immediately upstream and downstream of the site.
(TIF)

**S8 Fig. Statistical analysis of local structural environment around PNGS.** A. Bar plot showing the average frequency of each amino acid type within 6Å of all PNGS, canonical sequon PNGS (**B**), and non-canonical sequon PNGS (**C**). Blue bars are frequencies for the input list of residues, orange bars are the average frequencies across all 1000 permutation samples (with error bars and significance measures as described in S5 Fig), and green bars are the frequency of that amino acid within rEtpA.
(TIF)

**S9 Fig. Site-specific glycosylation analysis of native EtpA.** A. Glycosylation profile for rEtpA reproduced from Fig 3. B. Glycosylation profile of native EtpA from ETEC strain H10407. Left axis is % occupancy and right axis is peptide count. Note the much lower peptide counts for native EtpA.
(TIF)

**S10 Fig. Germline gene IgGBlast and somatic hypermutation analysis by ARMADILLO.** IgGBlast results summaries for 1C08 and 1G05 heavy chains (HC) and kappa chains (KC) showing the top match for V, D, and J mouse germline genes. Next to the summary tables are the percent identity to the top matching genes along with the gene names and the total number of somatic hypermutations (SHM) away from the predicted unmutated common ancestor (UCA). Also shown are output plots from the program ARMADILLO with SHM sites shown and scored by their probability, with red being the least probable and green being the most probable.
(TIF)

**S11 Fig. Mutation of 1G05 epitope glycan sites impacts avidity.** Figure at top depicts relative location of asparagine (N) to alanine (A) mutations to the putative 1G05 epitope. A. mAb 1G05 exhibits decreased avidity for recombinant mutant EtpA with N to A substitutions within C-terminal repeat region at positions 849, 1077, and 1305. Avidity indices (AI) were determined by kinetic ELISA with and without addition of 8 M urea as the chaotropic agent. AI (%) = (Vmax with urea)/(Vmax without urea) and expressed as % of the wild type recombinant protein. Comparisons by Kruskal-Wallis (n = 8 technical replicates/group from 2 independent experiments) *** = 0.0003, ** = 0.0037, * = 0.03. B. Immunoblot recognition of wild type and mutant protein by 1C08 and 1G05. PAGE image (left) indicates protein loading and MW markers. C. Blood group A binding by wild type and mutant protein in kinetic ELISA assay.
(TIF)

**S12 Fig. EMPEM processing workflow.** Simplified negative stain EMPEM data processing workflow including a representative micrograph, 2D class averages, and particle counts at each step.
(TIF)

**S13 Fig. AlphaFold2 predictions for full-length EtpA. A**. AlphaFold2 top ranking model colored by pLDDT score (blue = high). **B**. Per-residue sequence coverage and identity. **C**. Per-residue pLDDT scores. **D**. PAE (predicted aligned error) matrix.
(TIF)

**S1 Table. Strains and plasmids.**
(PDF)

**S2 Table. Primers.**
(PDF)

**S3 Table. Cryo-EM data collection, refinement, and validation.**
(PDF)

## Author Contributions

**Conceptualization:** Zachary T. Berndsen, Pardeep Kumar, Andrew B. Ward, James M. Fleckenstein.

**Data curation:** Zachary T. Berndsen, James M. Fleckenstein.

**Formal analysis:** Zachary T. Berndsen, Marjahan Akhtar, James M. Fleckenstein.

**Funding acquisition:** James M. Fleckenstein.

**Investigation:** Zachary T. Berndsen, Marjahan Akhtar, Mahima Thapa, Tim J. Vickers, Aaron Schmitz, Jonathan L. Torres, Sabyasachi Baboo, Pardeep Kumar, Nazia Khatoon, Alaullah Sheikh, Melissa Hamrick, Jolene K. Diedrich, Salvador Martinez-Bartolome, Patrick T. Garrett, Jackson S. Turner, Renee M. Laird, Frédéric Poly, Chad K. Porter, Jeffrey Copps, James M. Fleckenstein.

**Methodology:** Zachary T. Berndsen, Tim J. Vickers, Aaron Schmitz, Sabyasachi Baboo, Pardeep Kumar, Alaullah Sheikh, Jackson S. Turner, Ali H. Ellebedy, James M. Fleckenstein.

**Project administration:** Ali H. Ellebedy, Andrew B. Ward, James M. Fleckenstein.

**Resources:** Zachary T. Berndsen, John R. Yates, III, Andrew B. Ward, James M. Fleckenstein.

**Supervision:** John R. Yates, III, Ali H. Ellebedy, Andrew B. Ward, James M. Fleckenstein.

**Validation:** James M. Fleckenstein.

**Writing – original draft:** Zachary T. Berndsen, Tim J. Vickers, Aaron Schmitz, Alaullah Sheikh, Andrew B. Ward, James M. Fleckenstein.

**Writing – review & editing:** Zachary T. Berndsen, Tim J. Vickers, James M. Fleckenstein.

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
