## [Decision Letter · Decision Letter 0]

29 Jul 2024

Dear Dr. Fleckenstein,

Thank you very much for submitting your manuscript "Repeat modules and N-linked glycans define structure and antigenicity of a critical enterotoxigenic *E. coli* adhesin" for consideration at PLOS Pathogens. As with all papers reviewed by the journal, your manuscript was reviewed by members of the editorial board and by several independent reviewers. The reviewers were all very enthusiastic about the work, noting that it brings much needed attention to an important topic. Based on the reviews, we are likely to accept this manuscript for publication, providing that you modify the manuscript according to the review recommendations. These are mostly changes in the text and data presentation that were suggested to improve clarity and to note some potential limitations of the study. 

Sincerely,

Matthew A Mulvey, Ph.D.

Academic Editor

PLOS Pathogens

David Skurnik

Section Editor

PLOS Pathogens

Michael Malim

Editor-in-Chief

PLOS Pathogens

orcid.org/0000-0002-7699-2064

Reviewer Comments (if any, and for reference):

Reviewer's Responses to Questions

**Part I - Summary**

Reviewer #1: This manuscript on the structure, antigenicity, and potential of the secreted EtpA protein as a vaccine candidate against diarrheal disease caused by ETEC includes multiple experiments that advance the field.

Reviewer #2: This study is an important advancement in the understanding of ETEC EtpA structure and immunogenicity, especially regarding the roles of N-linked glycosylation, an increasingly recognized post-translational modification of a growing number of adhesins in a variety of gram-negative bacteria. The use of cryo-EM on EtpA-Fabs (from both monoclonal and polyclonal mouse and human sources) complexes revealed the location and specific residues involved in key EtpA-antibody interactions and enabled structural characterization of glycosylated EtpA. Mass spectrometry approaches uncovered the breadth of glycosylation of EtpA and revealed the impact of specific glycosylation sites on antibody binding were revealed.

Reviewer #3: The authors of this study use a combination of sophisticated structural biology, immunology, mass spectrometry, molecular biology, and ligand binding techniques in the service of investigating the highly conserved EtpA adhesin from enterotoxigenic E. coli. Although the crystal structure of EtpA had previously been solved, their cryo-EM structure in complex with two monoclonal antibodies, one that does and one that does not inhibit binding to Group A blood antigens on solid surfaces and tissue culture cells, provides new insights into the role of surface sugars in binding and suggest important epitopes that may be targeted with protective antibodies. The use of polyclonal serum from infected humans to further characterize binding sites using negative-stained EM and 2D particle analysis adds considerable confirmatory data of particular relevance to human health. The methods are well described, the results are convincing, and the conclusions are important and well-justified.

**Part II – Major Issues: Key Experiments Required for Acceptance**

Reviewer #1: (No Response)

Reviewer #2: Major points:

• Lines 290-293. The statement that there is “close agreement” between the rEtpA and native EtpA glycosylation patterns is difficult to believe given the comparison shown in Supp. Fig 7. These glycosylation profiles illustrated in Supp. Fig 7 look markedly different. Were other digestion enzymes (e.g, chymotrypsin) attempted to increase coverage of the protein and/or a larger scale purification attempted to increase yield? The apparent differences in the profiles need to be addressed.

• The recombinant expression system involving production of EptC from one plasmid and induction of EptA from another plasmid will most likely result in a scenario in which the molar ratio of EptC:EptA in the recombinant system differs significantly from the molar ratio of EptC:EptA in the native strain. This difference in molar ratio is likely to have significant impact on the glycosylation profiles in the two systems, a limitation should be acknowledged and addressed in the Discussion.

Reviewer #3: None noted

**Part III – Minor Issues: Editorial and Data Presentation Modifications**

Reviewer #1: The minor comments below need to be addressed. In addition, the large number of typographical errors in the submitted manuscript was distracting.

L 64 The structure presented is not at atomic structure, and the text should not state that it is.

L 91 is TpsA a synonym for EtpA? -- or is TpsA a class of proteins as implied on L 140? Your definition (L 88) seems to imply that Tps is the class (not TpsA), so your nomenclature is confusing

Fig 1B - what is 'membrane'? images have weird edges.

Fig 2 what portion of A is shown in B? what is the relative orientation of panels A and B? relative orientation of A and C? panel D, NTD vs NTS; NTD is only used here and one other place in the manuscript, and therefore it is confusing. How does this relate to the term N-terminal (TPS) domain, referred to in the legend of Fig 2B. -- NTD=NTS plus 65 N-term residues?

L 180-181 -- need to define an "extra-helical insert", because it appears that CTR4 has them too; you have a minimum length?

L 187-188 "potentially enhancing flexibility between the two domains". if this is true, then the cryoEM map should have limited resolution in this area, and this can be quantitatively shown. If the data don’t show this, then remove the statement from results, and put it in discussion

L 199 - H2 is not shown in Fig 2, so text is unclear

Fig 3 - is it the graphed line that is the peptide count? if so, the legend on the figure and in its text should indicate this.

panel D cross-section, label the faces. panel E - explain the BSA column in the table; a discussion of this result should be in the text of the manuscript, since this is such a large number compared to the rEtpA results

L 334 -define EMPEM

L 395-396 - finding 10 discrete polyclonal antibodies from the 600 amino acid NTD and 10 from the 900 amino acid CTR protein does not coincide with the statement that 'rEtpA possesses a large immunogenic surface'. Polyclonal antibodies are often seen decorating the entire surface of a protein, appearing as a fuzzy coat; the data do not support your statement

Reviewer #2: Minor points:

• Figure 1G: The meaning of the blue vs open circle data points is unclear.

• Figure 3D: The text in lines 262-263 indicates that panel 3D is a cryo-EM map, but the figure legend indicates that it is a structure with modeled N-linked glucose residues added. Please clarify.

• Many of the figures are very small, with labels and annotation that will be very difficult to read in the published paper. Examples include Fig 1 (especially 1A), Fig 3, Suppl Fig 2 (especially 2B), Suppl Fig 3 (especially Fig 3B), Suppl Fig 4 (especially Fig 4D), Suppl Fig 5, Suppl Fig 11 (especially Fig 11B, 11C, and 11D).

Reviewer #3: Humans are the only species that can volunteer, so the authors should search for every use of "human volunteers" and eliminate this redundancy.

Severity appears twice in the sentence spanning lines 38-40.

The word "to" should be deleted from line 70.

In line 253 "that" should be "than".

Where is the cartoon representation to which line 844 refers?

PLOS authors have the option to publish the peer review history of their article (what does this mean?). If published, this will include your full peer review and any attached files.

Reviewer #1: No

Reviewer #2: No

Reviewer #3: **Yes: **Michael Donnenberg

Figure Files:

Data Requirements:

Reproducibility:

References:

---

## [Editor Report · Decision Letter 1]

12 Aug 2024

Dear Dr. Fleckenstein,

We are pleased to inform you that your manuscript 'Repeat modules and N-linked glycans define structure and antigenicity of a critical enterotoxigenic *E. coli* adhesin' has been provisionally accepted for publication in PLOS Pathogens.

Best regards,

Matthew A Mulvey, Ph.D.

Academic Editor

PLOS Pathogens

David Skurnik

Section Editor

PLOS Pathogens

Michael Malim

Editor-in-Chief

PLOS Pathogens

orcid.org/0000-0002-7699-2064
---

## [Editor Report · Acceptance letter]

12 Sep 2024

Dear Dr. Fleckenstein,

We are delighted to inform you that your manuscript, "Repeat modules and N-linked glycans define structure and antigenicity of a critical enterotoxigenic *E. coli* adhesin," has been formally accepted for publication in PLOS Pathogens.

Best regards,

Michael Malim

Editor-in-Chief

PLOS Pathogens

orcid.org/0000-0002-7699-2064